# Smartphone Addiction and Sleep Quality on Academic Performance of University Students: An Exploratory Research

**DOI:** 10.3390/ijerph18168291

**Published:** 2021-08-05

**Authors:** Balan Rathakrishnan, Soon Singh Bikar Singh, Mohammad Rahim Kamaluddin, Azizi Yahaya, Mohd Azrin Mohd Nasir, Fauziah Ibrahim, Zaizul Ab Rahman

**Affiliations:** 1Faculty of Psychology and Education, University Malaysia Sabah, Kota Kinabalu 88400, Malaysia; soonbs@ums.edu.my (S.S.B.S.); azizi.yahaya@ums.edu.my (A.Y.); 2Centre for Research in Psychology and Human Well-Being, Faculty of Social Sciences and Humanities, University Kebangsaan Malaysia, Bangi 43600, Malaysia; rahimk@ukm.edu.my (M.R.K.); ifauziah@ukm.edu.my (F.I.); 3School of Applied Psychology, Social Work & Policy, UUM College of Arts & Sciences, University Utara Malaysia, Sintok 06010, Malaysia; mohdazrin@uum.edu.my; 4Research Centre for Theology and Philosophy, Faculty of Islamic Studies, University Kebangsaan Malaysia, Bangi 43600, Malaysia; zaizul@ukm.edu.my

**Keywords:** smartphone addiction, sleep quality, academic performance, health, university student

## Abstract

This study was conducted to examine the relationship between smartphone addiction, sleep quality, and academic performance. The study presented quantitative research on 323 students in a public university in Sabah to explore the relationship between smartphone addiction, sleep quality, and academic performance. A simple random sampling was used in the study. The Smartphone Addiction Scale Short Version (SAS-SV) and the Pittsburgh Sleep Quality Index (PSQI) were used in this study. SPSS was used as a tool of analysis for descriptive and inferential analysis. Pearson correlation was involved to test the hypothesis of the study. The result indicated that the greater the smartphone addiction, the lower the academic performance of university students. The finding also proved that students with poor sleep quality might exhibit low academic performance. Smartphone addiction was found to be associated with sleep quality where overusing smartphones was related to poor sleep quality in university students. On this basis, the problem of smartphone addiction and sleep quality should be tackled in order to improve the academic performance of university students and their overall health.

## 1. Introduction

Electronic devices have become popular across generations, with smartphones being identified as the most commonly used electronic device among the younger generation, especially among young adults [1]. Cohen reported that 50% of teenagers believe they are addicted to smartphones, whereas 59% of oldsters argue that their youngsters are strongly attracted by smartphones [2]. According to the survey, 45% of teenagers claim they use the internet “almost continuously,” up from 24% in 2014–2015 [3]. Therefore, it is evident that smartphone addiction has become a significant issue among adolescents. Smartphone addiction, a form of technology addiction, is defined as an impulse control disorder in which a person is exposed to the harmful effects of technology as a result of excessive use of the internet, video games, and mobile devices [4,5].

A previous study has also reported that smartphone addiction causes health deterioration due to decreased involvement in physical activities like walking and affects daily routine [6]. Smartphone addiction may additionally bring physical and psychological hurt that has shown within the past analysis and like depression, sleep disturbances, anxiety, relationship issues, digital eye strain, neck issues, and even automobile accidents because of the misuse of the smartphone such as texting or abuse of smartphone during driving [7,8,9].

The previous study also showed that the majority of students spend their time more on Facebook and other social media than on academic purposes [10]. Students who tend to send text messages and check Facebook while doing assignments were negatively related to their academic performance [11]. Thus, this has been proven that high addiction to smartphone addiction did hurt academic performance [12].

The study of metric weight unit by Ng, Hassan, Mohammad Nor and Abdul Malek suggested that smartphones negatively affect tutorial performance among students, even when these smartphones are used for school-related learning activities [13]. As students spend long periods of time on their smartphones, their participation in group activities and reading are significantly reduced [14]. In addition, smartphone addiction also affects students’ motivation to learn. Based on Lee, Cho, Kim, and Noh, the greater the addiction level, the less likely students are self-regulated to learn, and the lower the level of flow once students start learning [15]. Thus, it is crucial to review the connection between smartphone addiction and tutorial performance.

Sufficient sleep is important for students to improve their academic performance. Based on the report of Rafidi (2019), good sleep quality is adequate among university students [16]. There are several factors that affect sleep quality, such as stress from studying and the use of electronic gadgets during bedtime. Various studies worldwide have shown that insomnia is prevalent in 10–30% of the population, with some studies even recording prevalence as high as 50–60% [17]. Sleep quality has significant impacts on cognitive ability and physical strength, and the consequences of poor sleep quality are also serious, such as depression, impaired work performance, work-related motor vehicle accidents, and poor overall quality of life.

Besides smartphone addiction, sleep quality may also affect the educational performance of university students [18]. Poor sleep quality caused by excessive use of smartphones may affect the memory, decision-creating, learning, and concentration level of students [19,20]. This may result in students receiving low grades in their tutorials [21]. Several analyses have also suggested that poor sleep quality related to stress may bring negative impacts on tutorial performance [22,23]. Considering the findings of previous studies, we are interested to review the relationship between sleep quality and tutorial performance among university students.

A previous study by Soni, Upadhya, and Jain showed that smartphone dependence has a significant impact on sleep quality [24]. This study presented that poor sleep quality was more evident among heavy smartphone users [25]. Individuals who use smartphones at night are associated with poorer sleep quality, later wake-up time, and higher sleep latency [19]. Individuals who use smartphones immediately before sleep, or view their smartphone screens while trying to sleep, may experience reduced sleep efficiency and increased sleep latency [24].

This study consists of three main objectives. The first objective is to study the relationship between smartphone addiction and academic performance among students of a public university. The second objective is to study the relationship between sleep quality and academic performance among students of a public university. The third objective of the study is to study the relationship between smartphone addiction and sleep quality among students of a public university and how it relates to their overall health.

## 2. Literature Review

A recent study has shown that technology plays an important role in the aspects of contemporary life and exposes students to a great amount and variety of global information [26]. In the academic aspect, smartphones have become the devices for college students to send messages, submit assignments, and search for data [27]. However, the excessive use of smartphones or smartphone addiction may impact academic performance [12], as students tend to use their smartphones for entertainment purposes rather than for academic purposes [11].

Insufficient or poor sleep quality influences tutorial performance [28]. Students with lower sleep quality will exhibit poorer educational performance. Most of the students who perform poorly in exams seem to suffer from poor sleep quality. Multiple studies have indicated that insufficient sleep affects memory performance [29], decision-making, learning, and concentration level among students, hence resulting in bad academic performance [20].

Sleep quality is defined as the satisfaction of the sleep experience, integrating the aspects of sleep initiation, sleep maintenance, sleep quantity, and refreshment upon the awakening of an individual [30]. According to the National Sleep Foundation, good sleep quality for adults means a person falls asleep in 30 min or less, sleeps soundly through the night with no awakening, and drifts back to sleep within 20 min if you have awakened [31]. Sleep quality is also affected when the individual overuses a smartphone at bedtime [32].

Some reports suggest that internet use may affect sleep quality, leading to reduced rapid eye movement (REM) sleep, slow-wave sleep, and sleep efficiency [33]. The excessive use of smartphones and tablets in the evening causes difficulty to sleep [34] due to the blue light of smartphones which disrupts the melatonin production involved within the natural rhythms of our body [35]. Thus, sleep quality among individuals who use smartphones during bedtime will be affected.

## 3. Materials and Methods

The present research employed a quantitative research design using a survey research approach. For this, a self-administered questionnaire was used to collect data from university students. The questionnaire was comprised of two instruments namely the Smartphone Addiction Scale-Short Version (SAS-SV) and the Pittsburgh Sleep Quality Index (PSQI). The details of each instrument are as follows:

Smartphone Addiction Scale short version (SAS-SV): This instrument to examine smartphone addiction was developed by Kwon et al. [36]. This scale consisted of six factors (daily-life disturbance, positive anticipation, withdrawal, cyberspace-oriented relationship, overuse, and tolerance from the original version of the smartphone addiction scale) with ten items each, rated using the six-point Likert scale (1: strongly disagree, 2: disagree, 3: weakly disagree, 4: weakly agree, 5: agree, and 6: strongly agree) [36]. The scoring of SAS-SV was done by adding the scores altogether which yields a total score (numerical data). A cut-off value of 31 and 33 was advised for boys and girls, respectively; a higher score indicated a higher risk of addiction and a lower score indicated a lower risk of addiction [36]. The SAS-SV was considered valid and reliable with Cronbach’s alpha value of 0.911 [36].

Pittsburgh Sleep Quality Index (PSQI): The Pittsburgh Sleep Quality Index (PSQI) was developed by Buysse, Reynolds, Monk, Berman, and Kupfer [37]. According to Buysse et al. [37], the PSQI was used to provide a reliable, valid, and standardized measure of sleep quality, and to distinguish between good and poor sleepers among test takers [37]. The PSQI consisted of seven components (subjective sleep quality, sleep latency, sleep duration, habitual sleep efficiency, sleep disturbances, use of sleep mediation, and daytime dysfunction) with scores from 19 items. Each of the seven components was equally weighted on a scale ranging from 0 to 3. The total score was compiled and those respondents who scored 5 or less PSQI global score were classified as good sleepers, while higher scores represented poor sleepers. The PSQI showed internal consistency and a reliability coefficient (Cronbach’s alpha) of 0.83 [37].

A pilot study was carried out among 50 respondents from the late university student group of university students to ensure the reliability values of both instruments. The results of the pilot study indicated good reliability values for both instruments. SAS-SV showed a total reliability value of 0.77 (Cronbach’s alpha) while PSQI exhibited a total reliability of 0.70. Additionally, the demographic data, as well as Cumulative Grade Point of Average (CGPA) of participants, were collected.

In this study, the respondents were recruited using a simple random sampling technique as it helps to generalize the findings to the entire population. The entire study was obtained ethical approval; ref: 4KEtika 4/20(11)) from the Ethical Board of University Malaysia Sabah. Prior to data collection, signed consent was obtained from each respondent and they were assured with anonymity and confidentially of data. A total of 323 respondents were recruited after considering a dropout rate of 5%. The calculation of the total sample size was based on the total population of 1500 students and for this, the Krejcie and Morgan [33] sample size method was used. The samples were screened for a few selection criteria prior to the study. The selection criteria that were imposed were: from the age group of 20–27, currently active students, had completed at least one semester, and possessed a smartphone.

After successful data collection, all the responses were coded into the IBM Statistical Package for Social Sciences (version 25.0). This study uses a *p*-value, also known as a probability value, to indicate how likely it is the data happened by coincidence (i.e., that the null hypothesis is true). A statistically significant *p*-value is less than 0.05 (usually ≤0.05). It shows significant evidence against the null hypothesis since the null hypothesis has a less than 5% chance of being accurate (and the results are random). While *p*-value, which is greater than 0.05 (>0.05), shows strong support for the null hypothesis. This indicates that the null hypothesis is retained and the alternative hypothesis is rejected. Preliminary analyses were carried out to identify missing data as well as outliers. After the detailed process of data mining and cleaning, data normality of total scores of each variable was ensured using skewness and Kurtosis analyses. All data exhibited normal distribution and therefore parametric tests were carried out to achieve the objectives of this study.

## 4. Results

Based on Table 1, among the 323 respondents, 162 were female (50.20%) and 161 were male (49.84%). Based on Table 2, 84 respondents (26.00%) were Chinese, 70 respondents (21.67%) were Malay, 23 respondents (7.32%) were Indian, and 146 respondents (45.20%) were of other ethnicities including Bumiputera Sarawak, Bumiputera Sabah, Iban, Kadazan Dusun, Brunei, Kedayan, and others.

Table 3 shows that 179 respondents (55.42%) from the Faculty of Psychology and Education (FPP), 58 respondents (17.96%) from the Faculty of Business, Economics and Accounting (FPEP), 24 respondents (7.32%) from the Faculty of Engineering (FKJ), 21 respondents (6.50%) from Faculty of Humanities, Art and Heritage (FKSW), 20 respondents (6.19%) from Faculty of Science and Natural Resources (FSSA), 13 respondents (4.02%) from Faculty of Medicine and Health Sciences (FPSK), five respondents (1.55%) from Faculty of Food Science and Nutrition and three respondents (0.93%) from Faculty of Computing and Informatics.

From the aspect of age in Table 4, 133 respondents (41.18%) with age 23, 87 respondents (26.93%) with age 22, 45 respondents (13.93%) with age 21, 23 respondents (7.12%) with age 20, 23 respondents (7.12%) with age 24, five respondents (1.55%) with age 27, four respondents (0.93%) with age 26 and three respondents (0.93%) with age 25. From the aspect of the year of study in Table 5, there are 196 (60.68%) were year 3 students, 68 (21.05%) were year 1 student, 42 (13.0%) were year 2 students, nine (2.79%) were year 4 students and others were 8 (2.48) students.

### 4.1. Reliability of Instrument for the Pilot Study

The Smartphone Addiction Scale Short Version has a Cronbach’s Alpha of 0.77, indicating that the reliability of the questionnaire is excellent reliable. The Pittsburgh Sleep Quality Index has a Cronbach’s Alpha of 0.70, indicating the questionnaire is reliable (refer to Table 6).

### 4.2. Validity of Instrument for Pilot Study

The result of convergent validity has been shown in Table 7. The result of correlation shows that six of the components of the PSQI are significantly positively correlated with the Global PSQI with the (r = 0.45 to 0.76, *p* < 0.01), except component 6. According to Table 7, component 1 showed the strongest positive correlation with the Global PSQI among the seven components (r = 0.76, *p* < 0.01), component 7 showed the weaker correlation with the Global PSQI with (r = 0.45, *p* < 0.01). On the other hand, the correlation of the Global PSQI showed no correlation with total SAS-SV. Therefore, we indicate the PSQI is adequate for discriminant validity. In order words, these two were measuring different constructs.

### 4.3. Smartphone Addiction Scale Short Version

Table 8 showed the mean, standard deviation, and percentage of SAS-SV. The result shows that the respondent question number 1 (missing planned work due to smartphone use), number 2 (having a hard time concentrating in class, while doing an assignment, or working due to smartphone use), and number 9 (using my smartphone longer than I intended) of SAS-SV has the higher mean (M = 4.24, SD = 1.42), (M = 4.57, SD = 1.26), (M = 4.73, SD = 1.17). The result showed that most of the respondents having the problem of using my smartphone longer than I intended having a hard time concentrating in class, while doing the assignment, or working due to smartphone use ‘and missing planned work due to smartphone use’.

### 4.4. Pittsburgh Sleep Quality Index (PSQI)

Table 9 showed the result of mean, standard deviation and percentage of PSQI. According to table component 1 (subjective sleep quality), component 2 (Sleep latency), component 5 (sleep disturbance), component 7 (daytime dysfunction) has the higher mean and percentage indicated that most of the respondents has the problem of subjective sleep quality, sleep latency sleep disturbance and daytime dysfunction (M = 1.37, SD = 0.86), (M = 1.64, SD = 1.06), (M = 1.37, SD = 0.65), (M = 1.13, SD = 0.81). Component 6 (use of sleep medication) showed the least mean and percentage, which indicated most respondents did not agree with the use of sleep medication (M = 0.21, SD = 0.55).

### 4.5. There Is a Significant Relationship between Smartphone Addiction and Academic Performance

The correlation between smartphone addiction and academic performance is shown in Table 10. According to the results, there was a negative correlation between smartphone addiction and academic performance (r = −0.34, *p* < 0.05). The correlation between smartphone addiction and academic performance indicated that when the smartphone addiction of respondents increased, their academic performance decreased.

### 4.6. There Is a Significant Relationship between Sleep Quality and Academic Performance

Table 11 shows the correlation between sleep quality and academic performance. The results showed a negative correlation between sleep quality and academic performance (r = −0.44, *p* < 0.05). The negative correlation showed that when Global PSQI increased (poor sleep quality), the CGPA decreased.

### 4.7. There Is a Significant Relationship between Smartphone Addiction and Sleep Quality

Table 12 shows the correlation between smartphone addiction and sleep quality. Smartphone addiction and sleep quality showed a positive correlation (r = 0.49, *p* < 0.05), indicating that when smartphone addiction increased, the respondents had poorer sleep quality, and the Global PSQI increased.

## 5. Discussion

### 5.1. Smartphone Addiction and Academic Performance

The first objective is to evaluate the relationship between smartphone addiction and academic performance. The results indicated a significant relationship between smartphone addiction and academic performance. Thus, we concluded that the higher the smartphone addiction, the lower the academic performance. This finding is similar to several past studies.

The Uses and Gratifications Theory states that people will seek the media when they get satisfaction from the media [38]. Smartphones have become very important for adolescents in their daily lives and their studies. They use smartphones to release stress, communicate with friends and family, and search for information. As smartphones fulfill their needs such as knowledge searching, gaming, and communicating with friends, these devices become important in their life. Thus, adolescents may become more dependent on smartphones, eventually leading to smartphone addiction, affecting their academic performance as they may start to ignore their studies.

Furthermore, the study of Khan, Khalid, and Iqbal showed that the amount of time spent on smartphones is associated with adolescents’ academic performance and time management skills [39]. Individuals with smartphone addiction may have poor time management skills as they spend most of their time on their smartphones and ignore the other important things in their daily life. Poor time management due to smartphone addiction may also be associated with the academic performance of adolescents. This is because phone addiction may lead to poor performance [40,41].

The study of Chaudhury and Tripathy that examined the relationship between smartphone addiction and academic performance [12] concluded that high addiction to smartphones lowers academic performance. The study also found a strong relationship between internet connectivity and smartphone addiction. Adolescents with 4G or 3G internet connectivity on their smartphones may have higher risks of being addicted to their smartphones, leading to poor academic performance.

There is a significant correlation between smartphone addiction and academic performance. The study of Gladius, Sowmiya, Vidya, Archana, and William found that nighttime phone usage and time spent on mobile phones resulted in declined study habits, difficulty in concentration, an increase in missed classes, and going late for classes [42]. As smartphone addiction is associated with adolescents overusing their smartphones even at night, they may experience reduced sleeping hours, related strongly with their concentration the next day, especially in classes. When they cannot concentrate, they may learn nothing in the class, and this will be related to their exam results.

### 5.2. Relationship between Sleep Quality and Academic Performance

The results indicated that there was a significant relationship between sleep quality and academic performance. The study found that the poorer the sleep quality, the lower the academic performance.

Based on the past study of Becker, Adams, Orr, and Quilter, sleep quality is correlated to academic performance [43]. Good sleep is more likely associated with good physical health, which is the predictor for academic performance. the findings also reported that good sleepers may experience lower anxiety, better body health, and may feel more energized when studying. Poor sleep quality may also significantly relate to stress levels [23]. Good physical health and reduced stress, related to good sleep quality, may increase the motivation to study and improve academic performance [44,45].

Based on Maheshwari and Shaukat’s studies, there is a significant association between sleep quality and academic performance [21]. The result is similar to the study of Adelantado-Renau, Diez-Fernandez, Beltran-Valls, Soriano-Maldonado, and Moliner-Urdiales, which also showed that sleep quality is associated with academic performance [46]. This study states that poor sleep quality reduces daytime alertness and memory performance which is related to their attention and academic performance impairment.

A past study performed by Aung, Nurumal, and Zainal with 105 nursing adolescents in the International Islamic University Malaysia (IIUM) also showed a similar result [47]. The majority of the adolescents experienced poor sleep quality. The finding also proved that poor sleep quality is associated with low academic performance. Adolescents with good sleep quality have better results than adolescents with bad sleep quality.

### 5.3. Relationship between Smartphone Addiction and Overall Health

From the result, smartphone addiction exhibited a relationship with sleep quality. High levels of smartphone addiction resulted in higher PSQI scores, which represented poorer sleep quality.

Sleep quality is important for everyone. Good sleep can keep the heart healthy, prevent cancer, reduce stress, improve memory, and help weight loss [48]. However, nowadays, people stay up late for several reasons, affecting sleep quality. There are several reasons which cause poor sleep quality. Smartphone addiction is one of the important factors which cause people to stay up late and affect sleep quality.

Numerous studies have concluded that smartphones have associated with sleep quality. The use of smart devices such as smartphones and smart portable television for an extended period of time have been associated with sleeping and waking time, deteriorating health and disturbing daily life [49,50,51,52].

Several studies reported that the blue light of smartphones is associated with sleep quality. Based on Swiner, this blue light may affect sleep [53]. When the body is exposed to blue light at nighttime, the body’s biological clock is affected. This is because the blue light from the smartphone inhibits the brain to produce melatonin which helps us fall asleep [54]. Thus, we can conclude that the amount of screen time we are exposed to will associated our sleep quality. Adolescents who use their smartphone during their bedtime will have poor sleep quality.

## 6. Conclusions

This study has brought an important and beneficial effect to lecturers, parents, and university students. The study has showcased the negative associated of smartphone addiction and poor sleep quality of adolescents on their academic performance. The study has also suggested that smartphone addiction is not only associated with academic performance for adolescents but also with the sleep quality of university students. Thus, it is extremely important to take precautionary steps against smartphone addiction.

The result of the study will benefit the government, especially the Ministry of Education of Malaysia. When the ministry obtains knowledge on the association between smartphone addiction, sleep quality, and academic performances among adolescents, they can plan and create the best strategy to improve the academic performance of adolescents.

## Figures and Tables

**Table 1 ijerph-18-08291-t001:** Frequency and percentage of respondent gender.

Gender (*n* = 323)	Frequency (F)	Percent (%)
Male	161	49.85
Female	162	50.15
Total	323	100.0

**Table 2 ijerph-18-08291-t002:** Frequency and percentage of respondent ethnicity.

Ethnicity (*n* = 323)	Frequency (F)	Percent (%)
Chinese	84	26.01
Malay	70	21.67
Indian	23	7.12
Others	146	45.20
Total	323	100.0

**Table 3 ijerph-18-08291-t003:** Frequency and percentage of respondent faculty.

Faculty (*n* = 323)	Frequency (F)	Percent (%)
FPEP	58	17.96
FKI	3	0.93
FKJ	24	7.43
FSMP	5	1.55
FKSW	21	6.50
FPSK	13	4.02
FPP	179	55.42
FSSA	20	6.19
Total	323	100.0

**Table 4 ijerph-18-08291-t004:** Frequency and percentage of respondent age.

Age (*n* = 323)	Frequency (F)	Percent (%)
20	23	7.12
21	45	13.93
22	87	26.93
23	133	41.18
24	23	7.12
25	3	0.93
26	4	1.24
27	5	1.55
Total	323	100.0

**Table 5 ijerph-18-08291-t005:** Frequency and percentage of respondent’s year of study.

Year (*n* = 323)	Frequency (F)	Percent (%)
Year 1	68	21.05
Year 2	42	13.00
Year 3	196	60.68
Year 4	9	2.79
Other	8	2.48
Total	323	100.0

**Table 6 ijerph-18-08291-t006:** Reliability test of two scales.

Variable	Items	Cronbach’s Alpha Value
SAS SV	10	0.77
PSQI	7	0.70

**Table 7 ijerph-18-08291-t007:** Convergent validity and discriminant validity of the pilot study.

	Pearson Correlation								
	1	2	3	4	5	6	7	Global PSQI	Total SAS SV
Component 1									
Component 2	0.59 **								
Component 3	0.33	0.33							
Component 4	0.29	0.12	0.69 **						
Component 5	0.49 **	0.47 **	0.22	0.24					
Component 6	−0.08	−0.24	0.11	−0.11	−0.08				
Component 7	0.26	0.27	0.18	0.02	0.35	0.04			
Global PSQI	0.76 **	0.69 **	0.74 **	0.64 **	0.64 **	−0.06	0.45 **		
Total SAS	0.42 *	0.21	0.13	0.05	0.23	−0.46 *	−0.21	0.18	-

* *p* < 0.05 ** *p* < 0.01.

**Table 8 ijerph-18-08291-t008:** Mean and standard deviation of SAS-SV.

	Mean	Std. Deviation
SAS1	4.24	1.42
SAS2	4.57	1.26
SAS3	3.78	1.63
SAS4	3.81	1.64
SAS5	3.39	1.56
SAS6	3.23	1.52
SAS7	3.81	1.47
SAS8	3.82	1.65
SAS9	4.73	1.17
SAS10	3.20	1.64
Total SAS		

**Table 9 ijerph-18-08291-t009:** Mean, standard deviation, and percentage deviation of PSQI.

	Mean	Std. Deviation
Component 1	1.37	0.86
Component 2	1.64	1.06
Component 3	0.58	0.92
Component 4	0.58	0.87
Component 5	1.37	0.65
Component 6	0.21	0.55
Component 7	1.13	0.81
Global PSQI		

**Table 10 ijerph-18-08291-t010:** Pearson correlation between smartphone addiction and academic performance.

	Pearson Correlation	
	SAS-SV	CGPA
SAS-SV	-	
CGPA	−0.34 *	-

* *p* < 0.5.

**Table 11 ijerph-18-08291-t011:** Pearson correlation of sleep quality and academic performance.

	Pearson Correlation	
	Global PSQI	CGPA
Global PSQI	-	
CGPA	−0.44 *	-

* *p* < 0.5.

**Table 12 ijerph-18-08291-t012:** Pearson correlation of smartphone addiction and sleep quality.

	Pearson Correlation	
	SAS-SV	Global PSQI
SAS-SV	-	
Global PSQI	0.49 *	-

* *p* < 0.5.

## Data Availability

The data is available upon request from the corresponding authors.

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
