# Peer review of "Smartphone Addiction and Sleep Quality on Academic Performance of University Students: An Exploratory Research"

_ijerph, 2021, doi:10.3390/ijerph18168291_

Round 1
Reviewer 1 Report
Dear Authors,
Thank you for your amendments and comments. I'm happy to accept the manuscript with the current changes. Best wishes.
Author Response
Thank you so much
Reviewer 2 Report
- At line-35, going online multiple times daily and using phone at dining table doesn’t necessarily lead to any issue.
- Line-44, the word studying is not commonly used.
- Line-131, using “fall asleep in 30 minutes” or “Sleep Onset Latency (SOL) less than 30 minutes” instead of “go to sleep” would be more accurate.
- Where is subtitle 2?
- Line-244, what’s the percentage of SAS-SV? Also, if the mean in table-8 refers to averaged scores, and each responder select one of the six options, the observation could be SAS1,2,9 are the most severe problems, instead of most people having these issue. The mean score shows averaged level of severity, but not the number of individuals.
- Table-9 doesn’t include percentage of PSQI as stated at line-256.
- Line-266 and line-274, “significantly low negative correlation” is an odd statement. For words “significantly low”, does it mean almost no relationship? For the correlation coefficient r, either 0.34 or 0.44 doesn’t show strong association. But the p-values do show the results have statistical significance. These two concepts are different. As a result, the words “significant relationship” (i.e. line-275) may misleading the reader. Also, the thumb up standards (i.e. r>0.5 or p<0.05) are not mentioned in the method session.
- Title for table-12 is incorrect.
- Overall, the findings of this observation study could not be used as the direct evidence for the root cause of low academic performance. For example, the negative correlation between phone addiction and academic performance can be interpreted as phone addiction leads to low GPA or poor performance leads to phone addiction. Thus, at line-300, the statement about phone addiction affecting academic performance is weak.
Author Response
Hello dear Author
I have done the changes as required. I really hope this changes will be accepted.
Thnak you so much.
Regards
Balan Rathakrishnan
Round 2
Reviewer 2 Report
Significantly improved from the previous version.
This manuscript is a resubmission of an earlier submission. The following is a list of the peer review reports and author responses from that submission.
Round 1
Reviewer 1 Report
Thank you for your hard work writing the thesis.
There are many papers on CGPA, smart addiction, and sleep quality.
The part about the statistical method ended with correlation analysis, and other control variables could not be included.
It is an important part to reveal that there is a causal relationship to this, and your paper is inadequate in research design and statistical methods to prove it.
I'm not even sure if the research data passed the IRB review.
I don't think it's suitable to accept the manuscript.
Reviewer 2 Report
Dear Authors,
This study aimed to investigate the relationship between smartphone addiction, poor sleep and academic performance. Findings from survey-based research showed there was low-moderate correlations between response to the SAS-SV, PSQI and CGPA.
The idea of the study will be of interest to those in the field and I feel this can add to the body of scientific knowledge. However, overall, I feel that major revisions in all aspects of the manuscript. Details are given below.
Abstract
The abstract gives a good overview of the research. You have included the purpose, methods and results. The abstract refers to levels of concentration, grade retention, dropout, and stress and depression but this does not appear to be reported in the manuscript.
Introduction
The introduction could benefit from being rewritten. It is not clear why there is a separate literature review section; this could be incorporated within the introduction as there paragraphs that already discuss the topic or theme.
Line 37 – I am not sure that ‘smitten’ is an appropriate choice of word. Perhaps ‘strongly attracted’ is more appropriate, or use the words from the report ‘59 percent of their parents agree that their kids are addicted’.
Line 47-52. I think this would flow better as a paragraph solely focussed on smartphone addiction and physical health. Moving references to academic performance to the next paragraph (lines 54-62) would improve the flow of the introduction.
Lines 73-81. Again, to improve the flow of the introduction, this paragraph would be better served if it were read before the previous paragraph (lines 64-71).
Materials and Methods
The Materials and Methods section could be improved somewhat. Although there is a statement at the end of the manuscript, it would be useful to also include a statement that you obtained ethical approval and give details of the ethical committee in the methods section. Please ensure that you also include details of the inclusion and exclusion criteria.
The methods section does not include any details of how you have processed and analysed the data, including how you handled or excluded any missing data. While there is some debate on whether Pearson’s correlation should be used for data involving Likert scales (ordinal data), you have not provided details of how you have tested the assumptions including absence of outliers, the normality of variables, the linearity and homoscedasticity of the data.
You could provide details of when this survey was completed. Was it distributed during teaching semester, assessment period, during the holidays or throughout the year? Did the participants complete the survey at a convenient time?
Line 133. There appears to be some discrepancy between the value reported here, and the abstract/results. I’m assuming this is a typographical error.
Line 141. It is not clear what CGPA is. Please include the full term initially, then include the acronym subsequently.
Results
Much like the methods, the results section could be improved. There are some details that have been described in the methods but not in the results section. You could include details on age, year of study, faculty, and CGPA. You could include median values and interquartile range of each variable to give the reader an understanding of the outcomes.
The abstract refers to health (line 27) was this measured as a separate aspect in the survey?
Discussion
The discussion feels disjointed, and repetitive at times. The discussion should be revised to ensure that the discussion comprehensively covers a theme in a paragraph, then discusses another topic in a different paragraph.
Lines 191-204. Ensure that you are careful not to state cause-and-effect when interpreting your findings. Given that you assessed a relationship between smartphone addiction and academic performance, you should only state there is an association between the two variables.
Line 259-260. Given that you did not assess physical and mental health, I do not think you can substantiate these claims.